# SELFNORM AND CROSSNORM FOR OUT-OF-DISTRIBUTION ROBUSTNESS

## ABSTRACT

Normalization techniques are crucial in stabilizing and accelerating the training of deep neural networks. However, they are mainly designed for the independent and identically distributed (IID) data, not satisfying many real-world out-of-distribution (OOD) situations. Unlike most previous works, this paper presents two normalization methods, SelfNorm and CrossNorm, to promote OOD generalization. SelfNorm uses attention to recalibrate statistics (channel-wise mean and variance), while CrossNorm exchanges the statistics between feature maps. Self-Norm and CrossNorm can complement each other in OOD generalization, though exploring different directions in statistics usage. Extensive experiments on different domains (vision and language), tasks (classification and segmentation), and settings (supervised and semi-supervised) show their effectiveness.

## 1 INTRODUCTION

Normalization methods, e.g., Batch Normalization (Ioffe & Szegedy, 2015), Layer Normalization (Ba et al., 2016), and Instance Normalization (Ulyanov et al., 2016), play a pivotal role in training deep neural networks. Most of them try to make training more stable and convergence faster, assuming that training and test data come from the same distribution. However, few studies investigate normalization in improving OOD generalization in real-world scenarios. For example, image corruptions (Hendrycks & Dietterich, 2019), e.g., snow and blur, can cause test data out of the clean training distribution. Moreover, training on synthetic data (Richter et al., 2016) to generalize to realistic data can significantly reduce the annotation burden. This work aims to encourage the interaction between normalization and OOD generalization. Specifically, we manipulate feature mean and variance to make models generalize better to out-of-distribution data.

Our inspiration comes from the observation that channel-wise mean and variance of feature maps carry some style information. For instance, exchanging the RGB means and variances between two instances can transfer style between them, as shown in Figure 1 (a). For many tasks such as CIFAR classification (Krizhevsky et al., 2009), the style encoded by channel-wise mean and variance is usually less critical in recognizing the object than other information such as object shape. Therefore, we propose CrossNorm that swaps the channel-wise mean and variance of feature maps. CrossNorm can augment styles in training, making the model more robust to appearance changes.

Furthermore, given one image in different styles, we can reduce their style discrepancy if adjusting their RGB means and variances properly, as illustrated in Figure 1 (b). Intuitively, the style recalibration can reduce appearance variance, which may be useful in bridging distribution gaps between training and unforeseen testing data. To this end, we propose SelfNorm to use attention (Hu et al., 2018) to adjust channel-wise mean and variance automatically.

It is interesting to analyze the distinction and connection between CrossNorm and SelfNorm. At first glance, they take opposite actions (style augmentation v.s. style reduction). Even so, they use the same tool: channel-wise statistics and pursue the same goal: OOD robustness. Additionally, CrossNorm can increase the capacity of SelfNorm by style augmentation. SelfNorm, with the help from CrossNorm, can generalize better to OOD data.

**Concept and Intuition**. The style concept here refers to a family of weak cues associated with the semantic content of interest. For instance, the image style in object recognition can include many appearance-related factors such as color, contrast, and brightness. Style sometimes may help in

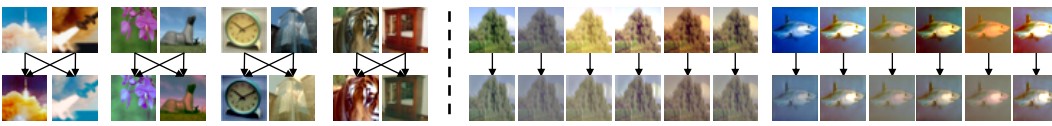

(a) Switch RGB mean and variance    (b) Recalibrate RGB mean and variance

Figure 1: CIFAR examples of exchanging (**Left**) and adjusting (**Right**) RGB mean and variance.

decision-making, but the model should weigh more on more vital content cues to become robust. To reduce its bias rather than discard it, we use CrossNorm with probability in training. The insight beneath CrossNorm is that each instance, or feature map, has its unique style. Further, style cues are not equally important. For example, the yellow color seems more useful than other style cues in recognizing orange. In light of this, the intuition behind SelfNorm is that attention may help emphasize essential styles and suppress trivial ones.

**Assumption**. Although we use the channel-wise mean and variance to modify styles, we do not assume that they are sufficient to represent all style cues. Better style representations are available with more complex statistics (Li et al., 2017) or even style transfer models (Ulyanov et al., 2017; Huang & Belongie, 2017). We choose the first and second-order statistics mainly because they are simple, efficient to compute, and can connect normalization to out-of-distribution generalization. In summary, the key contributions are:

- We propose SelfNorm and CrossNorm, two simple yet effective normalization techniques to enhance out-of-distribution generalization.
- SelfNorm and CrossNorm form a unity of opposites in using feature mean and variance for model robustness.
- They are domain agnostic and can advance state-of-the-art robustness performance for different domains (vision or language), settings (fully or semi-supervised), and tasks (classification and segmentation).

## 2 Related Work

**Out-of-distribution generalization**. Although the current deep models continue to break records on benchmark IID datasets, they still struggle to generalize to OOD data caused by common corruptions (Hendrycks & Dietterich, 2019) and dataset gaps (Richter et al., 2016). To improve the robustness against corruption, Stylized-ImageNet (Geirhos et al., 2019) conducts style augmentation to reduce the texture bias of CNNs. Compared to it, CrossNorm has two main advantages. First, CrossNorm is efficient as it transfer styles directly in the feature space of the target CNNs. However, Stylized-ImageNet relies on external style datasets and pre-trained style transfer models. Second, CrossNorm can advance the performance on both clean and corrupted data, while Stylized-ImageNet hurts clean generalization. In contrast to the consistent styles within one dataset, the external ones can result in massive distribution shifts. Recently, AugMix (Hendrycks et al., 2020c) trains robust models by mixing multiple augmented images based on random image primitives or image-to-image networks (Hendrycks et al., 2020a). Adversarial noises training (ANT) (Rusak et al., 2020) can also improve the robustness against corruption. CrossNorm is domain agnostic and orthogonal to AugMix and ANT, making it possible for their joint application. Moreover, unsupervised domain adaptation is also useful for corruption robustness in some situations (Schneider et al., 2020).

Besides common corruptions, generalization with distribution gaps (Richter et al., 2016) across different datasets also suffers from problems. IBN (Pan et al., 2018) mixes instance and batch normalization to shrink the domain distances. SelfNorm can bridge the domain gaps by style recalibration. Domain randomization (Yue et al., 2019) uses style augmentation for domain generalization on segmentation datasets. It suffers from the same issues of Stylized-ImageNet as it also uses pre-trained style transfer models and additional style datasets. By contrast, CrossNorm is more efficient and balances better between the source and target domains' performance. Beyond the vision field, many natural language processing (NLP) applications also face the out-of-distribution generalization challenges (Hendrycks et al., 2020b). Benefiting from the domain-agnostic property, SelfNorm and CrossNorm can also improve model robustness in the NLP area.

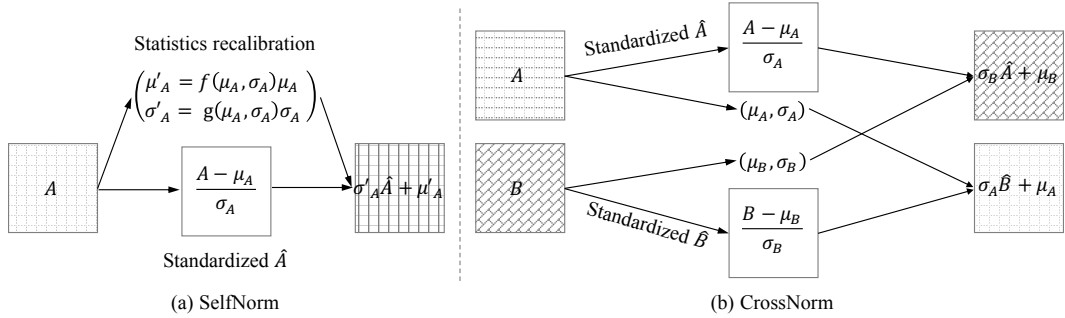

Figure 2: SelfNorm (**left**) and CrossNorm (**right**). SelfNorm uses attention to recalibrate the mean and variance of a feature map, while CrossNorm swaps the statistics between a pair of feature maps.

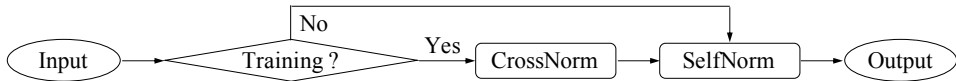

Figure 3: Flowchart for SelfNorm and CrossNorm. SelfNorm learns in training but functions in testing, while CrossNorm works in training.

**Normalization and attention**. Batch Normalization (Ioffe & Szegedy, 2015) is a milestone technique that inspires many following normalization methods such as Instance Normalization (Ulyanov et al., 2016), Layer Normalization (Ba et al., 2016), and Group Normalization (Wu & He, 2018). Recently, some works integrate attention (Hu et al., 2018) into feature normalization. Mode normalization (Deecke et al., 2018) and attentive normalization (Li et al., 2019) use attention to weigh a mixture of batch normalizations. Examplar normalization (Zhang et al., 2020) learns to combine multi-type normalizations by attention. By contrast, SelfNorm uses attention with only instance normalization. More importantly, different from previous normalization approaches, SelfNorm and CrossNorm are to improve out-of-distribution generalization. In addition, SelfNorm is different from SE (Hu et al., 2018), though they use similar attention. First, SelfNorm learns to recalibrate channel-wise mean and variance instead of channel features in SE. Second, SE models the interdependency between channels, while SelfNorm deals with each channel independently. Also, a SelfNorm unit, with $O(n)$, is more lightweight than a SE one, of $O(n^2)$, where $n$ denotes the channel number. The difference analysis here also applies to the Channel Attention (Zhang et al., 2018), similar to SE.

**Data augmentation**. Data augmentation is an important tool in training deep models. Current popular data augmentation techniques are either label-preserving (Cubuk et al., 2019a; Lim et al., 2019; Ho et al., 2019) or label-perturbing (Zhang et al., 2017; Yun et al., 2019). The label-preserving methods usually rely on domain-specific image primitives, e.g., rotation and color, making them inflexible for tasks beyond the vision domain. The label-perturbing techniques mainly work for classification and may have trouble in broader applications, e.g., segmentation. CrossNorm, as a data augmentation method, is readily applicable to diverse domains (vision and language) and tasks (classification and segmentation). The goal of CrossNorm is to boost out-of-distribution generalization, which is also different from many former data augmentation methods.

## 3 SELFNORM AND CROSSNORM

**Background**. Technically, SelfNorm and CrossNorm share the same origin: instance normalization (Ulyanov et al., 2016). In 2D CNNs, each instance has $C$ feature maps of size $H \times W$. Given $\boldsymbol{A} \in \mathbb{R}^{H \times W}$, instance normalization first normalizes the feature map and then conducts affine transformation:

$$\gamma \frac{\boldsymbol{A} - \mu_{\boldsymbol{A}}}{\sigma_{\boldsymbol{A}}} + \beta, \tag{1}$$

where $\mu_{\boldsymbol{A}}$ and $\sigma_{\boldsymbol{A}}$ are the mean and standard deviation; $\gamma$ and $\beta$ denotes learnable affine parameters. As shown in Figure 1 and also pointed out by the style transfer practices (Dumoulin et al., 2016; Ulyanov et al., 2017; Huang & Belongie, 2017), $\mu_{\boldsymbol{A}}$ and $\sigma_{\boldsymbol{A}}$ can encode some style information.

**SelfNorm**. SelfNorm replaces $\beta$ and $\gamma$ with recalibrated mean $\mu'_{\boldsymbol{A}} = f(\mu_{\boldsymbol{A}}, \sigma_{\boldsymbol{A}})\mu_{\boldsymbol{A}}$ and standard deviation $\sigma'_{\boldsymbol{A}} = g(\mu_{\boldsymbol{A}}, \sigma_{\boldsymbol{A}})\sigma_{\boldsymbol{A}}$, as illustrated in Figure 2, where $f$ and $g$ are the attention functions. The adjusted channel becomes:

$$\sigma'_{\boldsymbol{A}} \frac{\boldsymbol{A} - \mu_{\boldsymbol{A}}}{\sigma_{\boldsymbol{A}}} + \mu'_{\boldsymbol{A}}. \tag{2}$$

As $f$ and $g$ learn to scale $\mu_{\boldsymbol{A}}$ and $\sigma_{\boldsymbol{A}}$ based on themselves, $\boldsymbol{A}$ *normalizes itself by self-gating, hence SelfNorm*. SelfNorm is inspired by the fact that attention can help the model emphasize informative features and suppress less useful ones. In terms of recalibrating $\mu_{\boldsymbol{A}}$ and $\sigma_{\boldsymbol{A}}$, SelfNorm expects to highlight the discriminative styles and understate trivial ones. In practice, we use a fully connected (FC) network to wrap attention functions $f$ and $g$. The architecture is efficient as its input and output are both two scalars. Since each channel has its independent statistics, SelfNorm recalibrates each channel separately using $C$ lightweight FC networks, hence the complexity of $O(C)$.

**CrossNorm**. CrossNorm exchanges $\mu_{\boldsymbol{A}}$ and $\sigma_{\boldsymbol{A}}$ of channel $A$ with $\mu_{\boldsymbol{B}}$ and $\sigma_{\boldsymbol{B}}$ of channel $B$, i.e., changing $\beta$ and $\gamma$ to each other's $\mu$ and $\sigma$, shown in Figure 2:

$$\sigma_{\boldsymbol{B}} \frac{\boldsymbol{A} - \mu_{\boldsymbol{A}}}{\sigma_{\boldsymbol{A}}} + \mu_{\boldsymbol{B}} \qquad \sigma_{\boldsymbol{A}} \frac{\boldsymbol{B} - \mu_{\boldsymbol{B}}}{\sigma_{\boldsymbol{B}}} + \mu_{\boldsymbol{A}}, \tag{3}$$

where $\boldsymbol{A}$ *and* $\boldsymbol{B}$ *seem to normalize each other, hence CrossNorm.* CrossNorm is motivated by the key observation that a target dataset, such as a classification dataset, has rich, though subtle, styles. *Specifically, each instance, or even every channel, has its unique style.* Exchanging the statistics can perform efficient style augmentation, reducing the style bias in decision-making. In mini-batch training, we turn on CrossNorm with some probability.

**Unity of Opposites**. SelfNorm and CrossNorm both start from instance normalization but head in opposite directions. SelfNorm recalibrates statistics to focus on only necessary styles, reducing standardized features (zero-mean and unit-variance) and statistics mixtures' diversity. In contrast, CrossNorm transfers statistics between channels, enriching the combinations of standardized features and statistics. They perform opposite operations mainly because they target at different stages. SelfNorm dedicates to style recalibration during testing, while CrossNorm functions only in training. Note that SelfNorm is a learnable module, requiring training to work. Figure 3 shows the flowchart of SelfNorm and CrossNorm. Additionally, SelfNorm helps make the model less sensitive to appearance changes, while CrossNorm aims to lessen the model's style bias. Despite these differences, they both can facilitate out-of-distribution generalization. Further, CrossNorm can boost the performance of SelfNorm because its style augmentation can prevent SelfNorm from overfitting to specific styles. Overall, the two seemingly opposed methods form a unity of using normalization statistics to advance out-of-distribution robustness.

### 3.1 CrossNorm Variants

The core idea of CrossNorm is to swap mean and variance between channel features. For 2D CNNs, given one instance $\mathbf{X}, \in R^{C \times H \times W}$, CrossNorm can exchange statistics between its $C$ channels:

$$\{(\boldsymbol{A}, \boldsymbol{B}) \in (\mathbf{X}_{i,:,:}, \mathbf{X}_{j,:,:}) \mid i \neq j, 0 < i, j < C\}, \tag{4}$$

where $\boldsymbol{A}$ and $\boldsymbol{B}$ refer to the channel pair in Equation 3. If two instances $\mathbf{X}, \mathbf{Y} \in R^{C \times H \times W}$ given, CrossNorm can swap statistics between their corresponding channels, i.e., $\boldsymbol{A}$ and $\boldsymbol{B}$ become:

$$\{(\boldsymbol{A}, \boldsymbol{B}) \in (\mathbf{X}_{i,:,:}, \mathbf{Y}_{i,:,:}) \mid 0 < i < C\}. \tag{5}$$

Compared with one-instance CrossNorm, the two-instance one tends to consider instance-level style instead of channel-level.

Moreover, distinct spatial regions probably have different mean and variance statistics. To promote the style diversity, we propose to crop regions for CrossNorm:

$$\{(\boldsymbol{A}, \boldsymbol{B}) \in (\mathrm{crop}(\boldsymbol{A}), \mathrm{crop}(\boldsymbol{B})) \mid r_{\mathrm{crop}} \geq t\} \tag{6}$$

where the crop function returns a square with area ratio $r$ no less than a threshold $t(0 < t \leq 1)$. The whole channel is a special case in cropping. There are three cropping choices: content only, style only, and both. For content cropping, we crop A only when we use its standardized feature map. In other words, no cropping applies to A when it provides its statistics to B. Cropping both means cropping A and B no matter we employ their standardized feature map or statistics. The cropping strategy can produce diverse styles for both the two-instance and one-instance CrossNorms.

## 3.2 MODULAR DESIGN

SelfNorm and CrossNorm can naturally work in the feature space, making it flexible to plug them into many network locations. Two questions come: how many units are necessary and where to place them? To simplify the questions, we turn to the modular design by embedding them into a network cell. For example, in ResNet (He et al., 2016), we put them into a residual module. The search space significantly shrinks for the limited positions in a residual module. We will investigate the position choices in experiments. The modular design allows using multiple SelfNorms and CrossNorms in a network. We will show in the ablation study that accumulated style recalibrations are helpful for model robustness. Since excessive style augmentations are harmful (Geirhos et al., 2019), we randomly turn on only some CrossNorms in a forward process. Random sampling encourages diverse augmentations even though the same data pass multiple times.

## 4 EXPERIMENT

We evaluate SelfNorm and CrossNorm on out-of-distribution data that arise from image corruptions and dataset differences. The evaluation uses not only supervised and semi-supervised settings but also image classification and segmentation tasks. In addition to the vision tasks, we also apply them to a NLP task. Due to limited space, we leave all ablation studies in the appendix.

**Image classification datasets.** We use benchmark datasets: CIFAR-10 (Krizhevsky et al., 2009), CIFAR-100, and ImageNet(Deng et al., 2009). To evaluate the model robustness against corruption, we use the datasets: CIFAR-10-C, CIFAR-100-C, and ImageNet-C (Hendrycks & Dietterich, 2019). These datasets are the original test data poisoned by 15 everyday image corruptions from 4 general types: noise, blur, weather, and digital. Each noise has 5 intensity levels when injected into images.

**Image segmentation datasets.** We further validate our method using a domain generalization setting, where the models are trained without any target domain data and tested on the unseen domain. We use the synthetic dataset Grand Theft Auto V (GTA5) (Richter et al., 2016) as the source domain and generalize to the real-world dataset Cityscapes (Cordts et al., 2016). GTA5 has the training, validation, and test divisions of 12,403, 6,382, and 6,181, more than those of 2,975, 500, and 1,525 from Cityscapes. Despite the differences, their pixel categories are compatible with each other, making it possible to evaluate models' generalization capability from one to another.

**Sentiment classification datasets.** Besides vision tasks, we demonstrate that our method can also work well on NLP tasks. We use the out-of-distribution (OOD) generalization setting in binary sentiment classification. The model is trained on IMDb dataset (Maas et al., 2011) and is tested on SST-2 testing dataset (Socher et al., 2013). The IMDb dataset collects highly polarized full-length lay movie reviews with 25,000 positive and 25,000 negative reviews. The SST-2 contains 9613 and 1821 reviews for training and testing, which is also a binary sentiment classification dataset but instead contains pithy expert movie reviews.

**Metric.** For image classification, we use test errors to measure the robustness. Given corruption type $c$ and severity $s$, let $E_s^c$ denote the test error. For CIFAR datasets, we use the average over 15 corruptions and 5 severities: $1/75 \sum_{c=1}^{15} \sum_{s=1}^{5} E_{c,s}$. In contrast, for ImageNet, we normalize the corruption errors by those of AlexNet (Krizhevsky et al., 2012): $1/15 \sum_{c=1}^{15} (\sum_{s=1}^{5} E_s^c / \sum_{s=1}^{5} E_{c,s}^{AlexNet})$. The above two metrics follow the convention (Hendrycks et al., 2020c) and are denoted as mean corruption errors (mCE) whether they are normalized or not. Different from classification, segmentation use the mean Intersection over Union (mIoU) over all categories as metric. For sentiment classification, we report accuracy as the metric.

**Hyper-parameters.** In the experiments, a SelfNorm unit uses one fully connected layer, followed by Batch Norm and a sigmoid layer. We put CrossNorm ahead of SelfNorm, and plug them into every cell in a network, e.g., each residual module in a ResNet. During training, we turn on only some CrossNorms with probability to avoid excessive data augmentation. Usually, we conduct a grid search on four combinations of active numbers (1, 2) and probability (0.25, 0.5). For CrossNorm with cropping, we sample the bounding box ratio uniformly and set the threshold $t = 0.1$.

Table 1: mCE (%) comparison of CIFAR-10-C and CIFAR-100-C. SelfNorm&CrossNorm (SNCN) obtains lower errors than most previous methods with different backbones. Albeit some higher errors than AugMix, it is totally domain agnostic without relying on the image primitives, e.g., rotation, in AugMix. As SNCN and AugMix are orthogonal, their joint usage brings new state-of-the-art results.

| CIFAR-10-C | Basic | Cutout | Mixup | CutMix | AutoAug | AdvTr. | AugMix | SNCN | SNCN+AugMix |
|---|---|---|---|---|---|---|---|---|---|
| AllConvNet | 30.8 | 32.9 | 24.6 | 31.3 | 29.2 | 28.1 | 15.0 | 17.2 | **11.8** |
| DenseNet | 30.7 | 32.1 | 24.6 | 33.5 | 26.6 | 27.6 | 12.7 | 18.5 | **10.4** |
| WideResNet | 26.9 | 26.8 | 22.3 | 27.1 | 23.9 | 26.2 | 11.2 | 16.9 | **9.9** |
| ResNeXt | 27.5 | 28.9 | 22.6 | 29.5 | 24.2 | 27.0 | 10.9 | 15.7 | **9.1** |
| Mean | 29.0 | 30.2 | 23.5 | 30.3 | 26.0 | 27.2 | 12.5 | 17.0 | **10.3** |
| CIFAR-100-C | Basic | Cutout | Mixup | CutMix | AutoAug | AdvTr. | AugMix | SNCN | SNCN+AugMix |
| AllConvNet | 56.4 | 56.8 | 53.4 | 56.0 | 55.1 | 56.0 | 42.7 | 42.8 | **36.8** |
| DenseNet | 59.3 | 59.6 | 55.4 | 59.2 | 53.9 | 55.2 | 39.6 | 48.5 | **37.0** |
| WideResNet | 53.3 | 53.5 | 50.4 | 52.9 | 49.6 | 55.1 | 35.9 | 43.7 | **33.4** |
| ResNeXt | 53.4 | 54.6 | 51.4 | 54.1 | 51.3 | 54.4 | 34.9 | 40.8 | **30.8** |
| Mean | 55.6 | 56.1 | 52.6 | 55.5 | 52.5 | 55.2 | 38.3 | 43.5 | **34.7** |

Table 2: Clean error and mCE (%) of ResNet50 trained 90 epochs on ImageNet. SNCN, using simple domain-agnostic statistics, achieves comparable performance as AugMix. Jointly applying SNCN with AugMix and IBN can produce the lowest clean and corruption errors.

| Aug. | Clean | Noise | | | Blur | | | | Weather | | | | Digital | | | | mCE |
|---|---|---|---|---|---|---|---|---|---|---|---|---|---|---|---|---|---|
| | | Gauss. | Shot | Impulse | Defocus | Glass | Motion | Zoom | Snow | Frost | Fog | Bright | Contrast | Elastic | Pixel | JPEG | |
| Standard | 23.9 | 79 | 80 | 82 | 82 | 90 | 84 | 80 | 86 | 81 | 75 | 65 | 79 | 91 | 77 | 80 | 80.6 |
| Patch Uniform | 24.5 | 67 | 68 | 70 | 74 | 83 | 81 | 77 | 80 | 74 | 75 | 62 | 77 | 84 | 71 | 71 | 74.3 |
| Random AA* | 23.6 | 70 | 71 | 72 | 80 | 86 | 82 | 81 | 81 | 77 | 72 | 61 | 75 | 88 | 73 | 72 | 76.1 |
| MaxBlur pool | 23.0 | 73 | 74 | 76 | 74 | 86 | 78 | 77 | 77 | 72 | 63 | 56 | 68 | 86 | 71 | 71 | 73.4 |
| SIN | 27.2 | 69 | 70 | 70 | 77 | 84 | 76 | 82 | 74 | 75 | 69 | 65 | 69 | 80 | 64 | 77 | 73.3 |
| AugMix* | 23.4 | 66 | 66 | 66 | 69 | 80 | 65 | 68 | 72 | 72 | 66 | 60 | 63 | 78 | 66 | 71 | 68.4 |
| SNCN | 23.3 | 66 | 67 | 65 | 77 | 89 | 76 | 80 | 72 | 72 | 67 | 59 | 47 | 83 | 62 | 72 | 70.4 |
| SNCN+AugMix | **22.3** | 61 | 62 | 60 | 70 | 77 | 62 | 68 | 62 | 65 | 63 | 55 | 43 | 73 | 55 | 66 | **62.8** |

## 4.1 ROBUSTNESS AGAINST UNSEEN CORRUPTIONS FOR IMAGE CLASSIFICATION

**Supervised training on CIFAR.** Following AugMix (Hendrycks et al., 2020c), we evaluate Self-Norm and CrossNorm with four different backbones: an All Convolutional Network (Springenberg et al., 2014), a DenseNet-BC (k = 12, d = 100) (Huang et al., 2017), a 40-2 Wide ResNet (Zagoruyko & Komodakis, 2016), and a ResNeXt-29 (32×4) (Xie et al., 2017). We also use the same hyper-parameters in the AugMix Github repository[1].

According to Table 1, SelfNorm and CrossNorm can decrease the mean error by ∼12% on both CIFAR-10-C and CIFAR-100-C, outperforming most previous approaches on robustness against unseen corruptions. One possible explanation is that the corruptions, as demonstrated in Figure 12, mainly change image textures. SelfNorm and CrossNorm, through style recalibration and augmentation, may help reduce the texture sensitivity and bias, making the classifiers more robust to unseen corruptions. Also, the domain-agnostic SelfNorm and CrossNorm are orthogonal to Aug-Mix, which relies on domain-specific operations. Their joint application can continue to lower the mCEs by 2.2% and 3.6% on top of AugMix.

**Supervised training on ImageNet.** Following the AugMix Github repository, we train a ResNet-50 for 90 epochs with weight decay 1e-4. The learning rate starts from 0.1, divided by 10 at epochs 30 and 60. Note that AugMix reports the results of 180 epochs in their paper. For a fair comparison, we also train it 90 epochs in our experiments. Besides, we also add Instance-batch normalization (IBN) (Pan et al., 2018) in the final combination with AugMix. It was initially designed for domain generalization but can also boost model robustness against corruption.

Table 2 gives the results on ImageNet. We can find that SelfNorm and CrossNorm can make the corruption error drop by 10.2%. The clean error also descends by 0.6% simultaneously. More-over, applying SelfNorm and CrossNorm on top of AugMix can significantly lower its clean and corruption errors by 1.1% and 5.6%, achieving state-of-the-art performance. IBN also makes some contributions here since it is complementary to other components.

---

[1]https://github.com/google-research/augmix

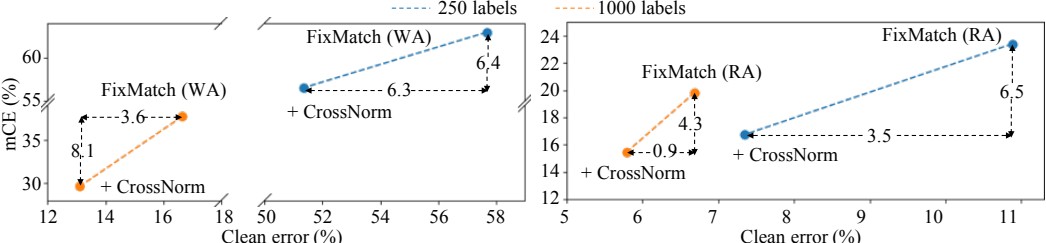

Figure 4: CrossNorm for semi-supervised CIFAR-10 classification. We apply CrossNorm on top of FixMatch with weak augmentation (WA) (**Left**), or strong RandAugment (RA) (**Right**). For either case, CrossNorm (CN) can substantially reduce both clean and corruption errors. Compared with RA, CrossNorm performs domain agnostic data augmentation, easily applicable to new domains.

**Semi-supervised training on CIFAR.** Apart from supervised training, we also evaluate CrossNorm in semi-supervised learning. Following state-of-the-art FixMatch (Sohn et al., 2020) setting, we train a 28-2 Wide ResNet for 1024 epochs on CIFAR-10. The SGD optimizer applies with Nesterov momentum 0.9, learning rate 0.03, and weight decay 5e-4. The probability threshold to generate pseudo-labels is 0.95, and the weight for unlabeled data loss is 1. We sample 250 and 4,000 labeled data with random seed 1, leaving the rest as unlabeled data. In each experiment, we apply CrossNorm to all data or only the unlabeled and choose the better one. Our experiments use the Pytorch FixMatch implementation [2], which has higher errors than the FixMatch reported. We choose it because it has the most stars among all the Pytorch implementations on Github.

Figure 4 shows the semi-supervised results. We run FixMatch with the strong RandAugment (Cubuk et al., 2019b) or only weak random flip and crop augmentations. With either FixMatch version, CrossNorm can always decrease both the clean and corruption errors, demonstrating its effectiveness in semi-supervised training. Especially, with the help of CrossNorm, training with 250 labels even has 3% lower corruption error than with 1000 labels, according to the right sub-figure. Additionally, two points are noteworthy here. First, we try FixMatch with only weak augmentations to simulate more general situations. For new domains other than natural images, humans may have the limited domain knowledge to design advanced augmentation operations. Fortunately, CrossNorm is domain-agnostic and easily applicable to such situations. Moreover, previous semi-supervised methods mainly focus on in-distribution generalization. Here we introduce out-of-distribution robustness as another metric for more comprehensive evaluation.

### 4.2 Generalization from Synthetic to realistic data for Image Segmentation

**Training setup.** We perform domain generalization from GTA5 (synthetic) to Cityscapes (realistic), following the setting of IBN (Pan et al., 2018). It uses 1/4 training data in GTA5 to match the data scale of Cityscapes. We train the FCN (Long et al., 2015) with ResNet50 backbone in source domain GTA5 for 80 epochs with batch size 16. The network is initialized with ImageNet pretrained weights. Then we test the model on both the source domain and target domain. The training uses random scaling, flip, rotation, and cropping ($713 \times 713$) for data augmentation. We use the 2-instance CrossNorm with style cropping in this setting. Besides, we re-implement the domain randomization (Yue et al., 2019) and make the training iterations the same as ours. It transfers the synthetic images to 15 auxiliary domains with ImageNet image styles.

**Results.** Based on Table 3, SelfNorm and CrossNorm both can substantially increase the segmentation accuracy on the target domain by 8.5% and 10.6%. SelfNorm learns to highlight the discriminative styles that are likely to share across domains. CrossNorm performs style augmentation to make the model focus more on domain-invariant features. SelfNorm and CrossNorm get comparable generalization performance as state-of-the-art IBN (Pan et al., 2018) and domain randomization (Yue et al., 2019). However, CrossNorm significantly outperforms the domain randomization method by 12.2% on the source accuracy. Because the domain randomization transfers external styles to the source training data, causing dramatic distribution shifts. Moreover, combining SelfNorm and CrossNorm gives the best generalization performance while still maintaining high source accuracy.

---

[2]https://github.com/kekmodel/FixMatch-pytorch

Table 3: Segmentation results (mIoU) on GTAV-Cityscapes domain generalization using a FCN with ResNet50. SelfNorm (SN) and CrossNorm (CN) are comparable with IBN and domain randomization (DR) on the target domain. Combining SN and CN can achieve state-of-the-art performance.

| Methods | Baseline | IBN | DR | SN | CN | SNCN |
|---------|----------|------|------|--------|------|--------|
| Source | 63.7 | 64.2 | 49.0 | **64.6** | 61.2 | 63.5 |
| Target | 21.4 | 29.6 | 32.7 | 29.9 | 32.0 | **36.5** |

Table 4: Accuracy (Acc) on OOD generalization for sentiment classification using GloVe embedding and ConvNets model. We train the model on IMDb source dataset and test on SST-2 target dataset.

| Methods | Baseline | SN | CN | SNCN |
|---------|----------|-----|-----|------|
| Source | 85.67 | **86.30** ($\uparrow$ **0.63**) | 85.14 ($\downarrow$ 0.53) | 85.92 ($\uparrow$ 0.25) |
| Target | 71.86 | 73.93 ($\uparrow$ 2.07) | 73.03 ($\uparrow$ 1.17) | **74.91** ($\uparrow$ **3.05**) |

### 4.3 OUT-OF-DISTRIBUTION GENERALIZATION FOR SENTIMENT CLASSIFICATION

**Setup.** We also conduct out-of-distribution generalization on the binary sentiment classification task in the NLP field to validate the versatility of SelfNorm and CrossNorm. The model is trained on the IMDb dataset and then tested on SST-2 dataset. Follow the setting of Hendrycks et al. (2020b), we use GloVe (Pennington et al., 2014) word embedding and the Convolutional Neural Networks (ConvNets) (Kim, 2014) as the classification model. We use the implementation of ConvNets in this repository[3]. The convolutional layers with three kernel sizes (3,4,5) are used to extract $n - gram$ features within the review texts. The SelfNorm and CrossNorm are placed between the embedding layer and the convolutional layers. We use the Adam optimizer and train the model for 20 epochs.

**Results.** From Table 4, we can find that SelfNorm improves the performance in both the source and target domains by 2.07% and 0.63%. CrossNorm can also increase target accuracy without much degradation in the source domain. Combining them gives a 3.05% accuracy boost. This experiment indicates that SelfNorm and CrossNorm can also work in the NLP area, not limited to the vision tasks. Despite the lack of intuitive explanations as for the image data, the mean and variance statistics in NLP data are also useful in facilitating out-of-distribution generalization.

### 4.4 VISUALIZATION

Apart from the quantitative comparisons, we also provide some visualization results of SelfNorm and CrossNorm to understand their effects better. It is nontrivial to visualize them directly in feature space. To deal with this, we map the feature changes made by SelfNorm and CrossNorm back to image space by inverting the feature representations (Mahendran & Vedaldi, 2015). For detailed experimental settings, refer to the appendix.

To visualize SelfNorm at a network location, we first forward an image to obtain the target representation immediately after the SelfNorm. Then we turn off the chosen SelfNorm and optimize the original image to make its representation fit the target one. In this way, we can examine SelfNorm's effect by observing changes in image space. As shown in Figure 6, SelfNorm can primarily reduce the contrast and color at the first network block. The effect becomes more subtle as SelfNorm goes deeper into the network. One possible explanation is that the high-level representations lose many low-level details, making it difficult to visualize their changes.

In addition to visualizing individual SelfNorms, it is also interesting to see their compound effect. To this end, we reconstruct an image from random noises by matching its representation with a given one. The reconstructed image can show what information is preserved by the feature representation. By comparing two reconstructed images from a network with or without SelfNorm, we can observe the joined recalibration effects of SelfNorms before a selected location. From Figure 7, we can find SelfNorms in the first two network blocks can suppress much style information and preserve object shapes. The reconstructions from block 3 do not look visually informative due to the high-level abstraction. Even so, SelfNorms can restrain the high-frequency signals kept in the vanilla network.

---

[3]https://github.com/bentrevett/pytorch-sentiment-analysis

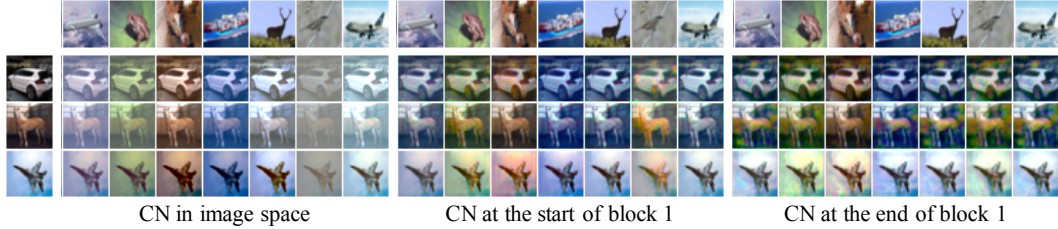

CN in image space        CN at the start of block 1        CN at the end of block 1

Figure 5: CrossNorm visualization at image level (**Left**), the head (**Middle**) and tail (**Right**) of block 1 in a WideResNet-40-2. Both the content (**Row**) and style (**Column**) images are from CIFAR-10. The style rendering changes from global to local as CrossNorm gets deeper in the network.

SN at the start of block 1 ¦ SN at the end of block 1 ¦ SN at the end of block 2 ¦ SN at the end of block 3

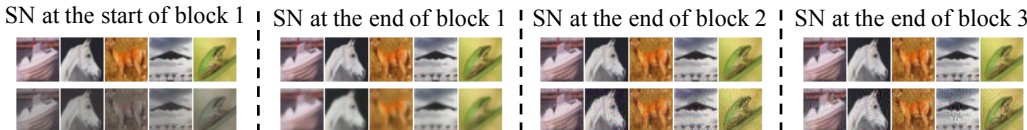

Figure 6: Visualizing 4 single SelfNorms by comparing images before (**Top**) and after (**Bottom**) them. The left two, lying in shallow locations, can adjust styles by suppressing color and adding blur. As SelfNorm goes deeper, the recalibration effect is subtle, due to the high-level feature abstraction.

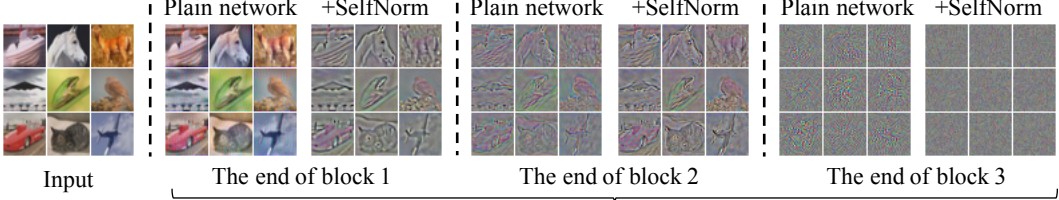

Input        The end of block 1        The end of block 2        The end of block 3

Reconstructed images from intermediate CNN features

Figure 7: Visualizing accumulated SelfNorms by comparing inverted images. SelfNorms in block 1 can wash away much style information preserved in the vanilla network. Similarly, the plain network's final representation retains some high-frequency signals which are suppressed by SelfNorms.

In the CrossNorm visualization, we pair one content image with multiple style images for better illustration. We first forward them to get their representations at a chosen position. Then, we compute the standardized features from the content image representation and the means and variances of the style image representation. The optimization starts from the content image and tries to fit its representation to the target one mixing the standardized features with different means and variances. Figure 5 shows diverse style changes made by CrossNorm. The style changes become more local and subtle as CrossNorm moves deeper in the network.

## 5    CONCLUSION

In this paper, we have presented SelfNorm and CrossNorm, two simple yet effective normalization techniques to improve OOD robustness. They form a unity of opposes as they confront and conform to each other in terms of approach (statistics usage) and goal (OOD robustness). Beyond their extensive applications, they may also shed light on developing domain agnostic methods applicable to multiple fields such as vision and language, and broad OOD generalization circumstances such as unseen corruptions and distribution gaps across datasets. Given the simplicity of SelfNorm and CrossNorm, we believe there is substantial room for improvement. The current channel-wise mean and variance are not optimal to encode diverse styles. One possible direction is to explore better style representations.

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

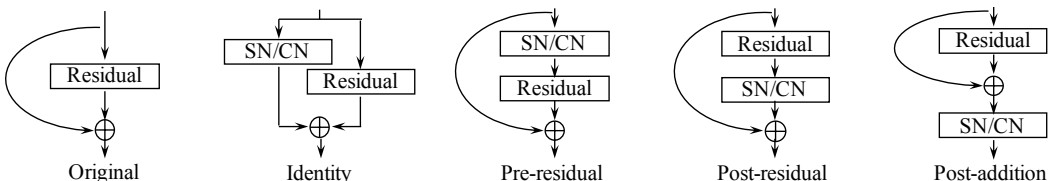

Figure 8: Illustration of SelfNorm (SN) and CrossNorm (CN) positions in a residual module.

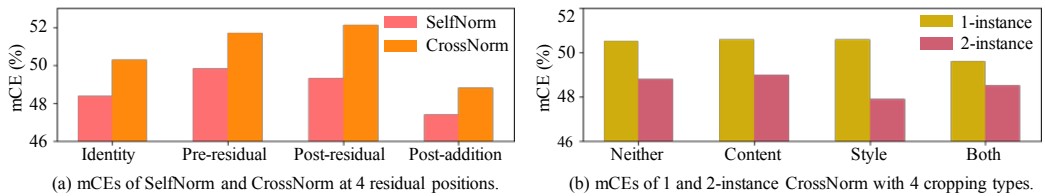

(a) mCEs of SelfNorm and CrossNorm at 4 residual positions.   (b) mCEs of 1 and 2-instance CrossNorm with 4 cropping types.

Figure 9: Study of Modular positions (**left**) and CrossNorm variants (**right**). SelfNorm and Cross-Norm both get the lowest errors at the post-addition position. Second, the 2-instance mode consistently outperforms the 1-instance one, and proper cropping can decrease the error.

## 6 SUPPLEMENTARY

### 6.1 ABLATION STUDY ON DESIGN CHOICES

All the experiments here use a 40-2 Wide ResNet, measured by the mCE on CIFAR-100-C.

**CrossNorm variants**. CrossNorm can be in 1-instance or 2-instance mode with four cropping options. Figure 9 compares their mCEs. The 2-instance mode constantly gets lower errors than the 1-instance. Furthermore, cropping can help decrease the error since it can encourage the style augmentation diversity. Note that style cropping may not always be superior. We find that the best cropping choice may change over backbones, datasets, and joint application with SelfNorm.

**Modular positions**. Here we investigate four positions in a residual module, illustrated in Figure 8. We can find that, in Figure 9, both SelfNorm and CrossNorm work the best at the post-addition position. For positions in other module types, refer to the ablation study in Section 6.2. Further, we find that their order seems to have little influence on the performance, as indicated in Table 5.

**Blocks choices for SelfNorm and CrossNorm**. After narrowing down their positions in a cell, we still need to study which blocks in a network should build on the cells with them. According to Table 6, They both perform the best when plugged into all blocks.

**Ablating components**. SelfNorm learns to recalibrate test styles to decrease texture sensitivity, while CrossNorm augments training styles to reduce texture bias. Although focusing on distinct aspects, they can lower the corruption error both separately and jointly, according to Table 7. On top of them, cropping and the consistency regularization in AugMix can further advance model robustness. Besides, SelfNorm outperforms SE (Hu et al., 2018) for model robustness because it takes advantage of the style information.

### 6.2 ABLATION STUDY ON CIFAR

In addition to the residual module, we also investigate the positions of SelfNorm and CrossNorm in the cells of AllConvNet, DenseNet, and ResNeXt. Figure 10 illustrates the positions in the All-ConvNet and DenseNet cells. The position results of SelfNorm and CrossNorm on two datasets CIFAR-10-C and CIFAR-100-C are given in Tables 8, 9, 10, and 11. We also report the results of SelfNorm and CrossNorm with different cropping choices in Tables 12 and 13. The full component ablation studies with different backbones are given in Tables 14 and 15.

Table 5: Order of SelfNorm and CrossNorm. In this experiment, they both are at the post-addition position in a residual cell.

| Order | SN→CN | CN→SN |
|---|---|---|
| mCE(%) | 46.9 | 46.6 |

Table 6: Block choices of SelfNorm and CrossNorm. We compare the mCEs (%) when applying SelfNorm and CrossNorm to image space or different blocks in a WideResNet-40-2. The choice of all blocks gives the lowest errors for both of them.

| Stages | N/A | Image | Block 1 | Block 2 | Block 3 | All |
|---|---|---|---|---|---|---|
| SelfNorm | 53.3 | 52.9 | 48.9 | 52.2 | 51.3 | **47.4** |
| CrossNorm | 53.3 | 54.3 | 52.2 | 51.2 | 51.5 | **48.8** |

Table 7: Ablation study of SE, SelfNorm(SN), CrossNorm(CN), cropping, and consistency regularization(CR). SN obtains much lower corruption error than SE, justifying its robustness superiority. Besides, SN and CN can work together in reducing mCE, and cropping and CR can help further.

| | Basic | SE | SE(post) | SN | CN | SNCN | SNCN+Crop | SNCN+Crop+CR |
|---|---|---|---|---|---|---|---|---|
| mCE(%) | 53.3 | 52.3 | 51.0 | 47.4 | 48.8 | 46.6 | 44.5 | **43.7** |

## 6.3 VISUALIZATION CONTINUED

**Visualization setup**. Our visualization builds on the technique: understanding deep image representations by inverting them (Mahendran & Vedaldi, 2015). The goal is to find an image whose feature representation best matches the given one. The search is done automatically by a SGD optimizer with learning rate 1e4, momentum 0.9, and 200 iterations. The learning rate is divided by 10 every 40 iterations. During the optimization, the network is in its evaluation mode with its parameters fixed. In the experiment, we use WideResNet-40-2 and images from CIFAR-10. In visualizing CrossNorm, we use the training images and a model trained for 1 epoch. The SelfNorm visualization uses test images and a well-trained model. We use different settings for them because CrossNorm is for training, while SelfNorm works in testing.

**More visualization results**. Figure 11, extending Figure 5, shows more CrossNorm visualizations in deeper network blocks. Figure 12 gives an illustration of 15 corruptions used in robustness evaluation on CIFAR and ImageNet. Moreover, Figure 13 shows some synthetic images from GTA5 and realistic ones from Cityscapes. The visualization of CrossNorm applied to the synthetic images is provided in Figure 14.

## 6.4 ABLATION STUDY ON IMAGENET

Table 16 reports the results of applying SelfNorm or CrossNorm with IBN. We can see that they can cooperate to improve corruption robustness. Moreover, in Tables 17 and 18, we also investigate the SelfNorm and CrossNorm positions in a residual module using ImageNet. Similar to the CIFAR results, the post-addition position performs the best for corruption robustness.

We also compare CrossNorm to Stylized-ImageNet, which transfers styles from external datasets to perform style augmentation. Stylized-ImageNet finetunes a pre-trained ResNet-50 for 45 epochs with double data (stylized and original ImageNets) in each epoch. To compare CrossNorm with Stylized-ImageNet, we perform the finetuning for 90 epochs using only the original ImageNet. In Table 19, although Stylized-ImageNet has 2% lower corruption error than CrossNorm, its clean error is 3.8% higher. Because the external styles in Stylized-ImageNet cause large distribution shifts, impairing its clean generalization. In contrast, The more consistent yet diverse internal styles help CrossNorm decreases both corruption and clean errors.

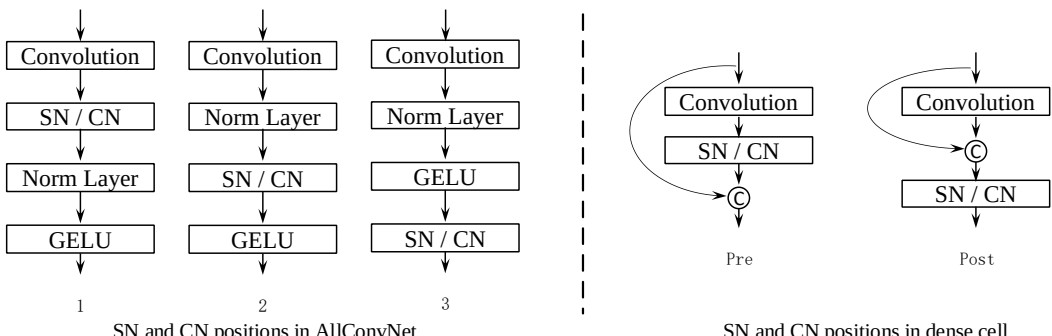

Figure 10: Illustration of SelfNorm (SN) and CrossNorm (CN) positions in AllConvNet block, and dense cells in DenseNet. For blocks in AllConvNet, we name the position after convolution layer as *1*, after normalization layer as *2*, and after GELU layer as *3*. For dense cells in DenseNet, we label the position before feature concatenation as *Pre*, and after concatenation as *Post*.

## 6.5 ABLATION STUDY ON GENERALIZATION FROM SYNTHETIC TO REALISTIC DATA

In addition to Table 3, Table 20 provides the ablation study on cropping for domain generalization from GTA5 to Cityscapes.

Table 8: Ablation study of the impact of different SelfNorm position for AllConvNet, DenseNet, WIdeResNet and ResNeXt on CIFAR-10-C measured by mCE

| Position | 1 | 2 | 3 | - |
|---|---|---|---|---|
| AllConvNet | **24.01** | 26.38 | 25.56 | - |
| Position | Pre | Post | - | - |
| DenseNet | 23.40 | **21.96** | - | - |
| Position | Residual | Post | Pre | Identity |
| WideResNet | 22.69 | 21.29 | **20.78** | 22.29 |
| Position | Residual | Post | Pre | Identity |
| ResNeXt | 21.94 | 24.76 | **21.49** | 21.99 |

Table 9: Ablation study of the impact of different CrossNorm position for AllConvNet, DenseNet, WIdeResNet and ResNeXt on CIFAR-10-C measured by mCE

| Position | 1 | 2 | 3 | - |
|---|---|---|---|---|
| AllConvNet | **25.99** | 26.27 | 26.84 | - |
| Position | Pre | Post | - | - |
| DenseNet | **24.72** | 29.17 | - | - |
| Position | Residual | Post | Pre | Identity |
| WideResNet | 25.20 | **21.62** | 24.91 | 23.27 |
| Position | Residual | Post | Pre | Identity |
| ResNeXt | 26.71 | **22.37** | 23.75 | 26.92 |

Table 10: Ablation study of the impact of different SelfNorm position for AllConvNet, DenseNet, WIdeResNet and ResNeXt on CIFAR-100-C measured by mCE

| Position | 1 | 2 | 3 | - |
|---|---|---|---|---|
| AllConvNet | **50.30** | 51.63 | 51.03 | - |
| Position | Pre | Post | - | - |
| DenseNet | **53.86** | 54.67 | - | - |
| Position | Residual | Post | Pre | Identity |
| WideResNet | 49.28 | **47.44** | 49.82 | 48.43 |
| Position | Residual | Post | Pre | Identity |
| ResNeXt | **47.56** | 49.00 | 50.86 | 50.44 |

Table 11: Ablation study of the impact of different CrossNorm position for AllConvNet, DenseNet, WIdeResNet and ResNeXt on CIFAR-100-C measured by mCE

| Position | 1 | 2 | 3 | - |
|---|---|---|---|---|
| AllConvNet | **52.20** | 52.49 | 52.69 | - |
| Position | Pre | Post | - | - |
| DenseNet | **55.44** | 57.57 | - | - |
| Position | Residual | Post | Pre | Identity |
| WideResNet | 52.06 | **48.76** | 51.74 | 50.26 |
| Position | Residual | Post | Pre | Identity |
| ResNeXt | 51.47 | **46.95** | 47.87 | 50.22 |

Table 12: Ablation study of the impact of SelfNorm + CrossNorm with different cropping, including neither, content, style and both, for AllConvNet, DenseNet, WIdeResNet and ResNeXt on CIFAR-10-C measured by mCE

| Backbone | Neither | Content | Style | Both |
|---|---|---|---|---|
| AllConvNet, 1 | 18.98 | 20.29 | **18.82** | 20.28 |
| DenseNet, Conv1 Pre | 18.75 | **18.23** | 18.70 | 18.83 |
| WideResNet, Post | 17.93 | 17.99 | **16.77** | 17.47 |
| ResNeXt, Post | **17.73** | 18.52 | 18.37 | 18.59 |

Table 13: Ablation study of the impact of SelfNorm + CrossNorm with different cropping, including neither, content, style and both, for AllConvNet, DenseNet, WIdeResNet and ResNeXt on CIFAR-100-C measured by mCE

| Backbone | Neither | Content | Style | Both |
|---|---|---|---|---|
| AllConvNet, 1 | 44.17 | 46.92 | **43.86** | 46.08 |
| DenseNet, Conv1 Pre | 51.38 | 49.40 | 49.13 | **48.97** |
| WideResNet, Post | 46.58 | 45.08 | 45.82 | **44.46** |
| ResNeXt, Post | **41.03** | 44.85 | 42.96 | 46.52 |

Table 14: Ablation study of each component of our methods, including SelfNorm, CrossNorm with cropping, consistency and combination with AugMix, for AllConvNet, DenseNet, WIdeResNet and ResNeXt on CIFAR-10-C measured by mCE

| Backbone | Basic | SN | CN | SN+CN +Crop | SN+CN+Crop +Consistency | AugMix | SNCN+Crop +AugMix |
|---|---|---|---|---|---|---|---|
| AllConvNet, 1, style | 30.80 | 24.01 | 25.99 | 18.82 | 17.24 | 15.00 | **11.79** |
| DenseNet, Conv1 Pre, both | 30.70 | 21.96 | 24.72 | 18.83 | 18.53 | 12.7 | **10.40** |
| WideResNet, Post, both | 26.90 | 20.78 | 21.62 | 17.47 | 16.93 | 11.2 | **9.94** |
| ResNeXt, Post, neither | 27.50 | 21.49 | 22.37 | 17.73 | 15.69 | 10.90 | **9.09** |

Table 15: Ablation study of each component of our methods, including SelfNorm, CrossNorm with cropping, consistency and combination with AugMix, for AllConvNet, DenseNet, WIdeResNet and ResNeXt on CIFAR-100-C measured by mCE

| Backbone | Basic | SN | CN | SN+CN +Crop | SN+CN+Crop +Consistency | AugMix | SNCN+Crop +AugMix |
|---|---|---|---|---|---|---|---|
| AllConvNet, 1, style | 56.40 | 50.30 | 52.20 | 43.86 | 42.83 | 42.7 | **36.80** |
| DenseNet, Conv1 Pre, both | 59.30 | 53.86 | 55.44 | 48.97 | 48.48 | 39.60 | **36.95** |
| WideResNet, Post, both | 53.30 | 47.44 | 48.76 | 44.46 | 43.70 | 35.90 | **33.38** |
| ResNeXt, Post, neither | 53.40 | 47.56 | 46.95 | 41.03 | 40.84 | 34.9 | **30.76** |

Table 16: Ablation study of IBN, SelfNorm(SN), CrossNorm(CN), consistency regularization(CR), and AugMix(AM) on ImageNet-C with ResNet50. IBN, originally designed for domain generalization, can also decrease mCE. Both SN and CN can further lower the error based on IBN. Combining them with AM gives the best robustness performance.

| | ResNet50 | ResNet50+IBN(a) | | | | ResNet50+IBN(b) | | | |
|---|---|---|---|---|---|---|---|---|---|
| | Basic | Basic | +CN | +CN+CR | +CN+AM | Basic | +SN | +SN+AM | +SNCN+AM |
| Clean err(%) | 23.9 | 23.2 | 23.1 | 22.6 | 22.5 | 24.0 | 23.5 | **22.3** | **22.3** |
| mCE(%) | 80.6 | 75.1 | 73.2 | 73.6 | 66.4 | 74.1 | 72.6 | 64.1 | **62.8** |

Table 17: Position investigation of SelfNorm in a residual module of ResNet50 trained 90 epochs on ImageNet.

| Modular Position | Identity | Pre-Residual | Post-Residual | Post-Addition |
|---|---|---|---|---|
| Clean error (%) | 24.0 | **23.0** | 23.2 | 23.7 |
| mCE(%) | 75.5 | 75.8 | 74.8 | **73.4** |

Table 18: Position investigation of CrossNorm in a residual module of ResNet50 trained 90 epochs on ImageNet.

| Modular Position | Identity | Pre-Residual | Post-Residual | Post-Addition |
|---|---|---|---|---|
| Clean error (%) | 25.2 | **23.4** | 23.5 | **23.4** |
| mCE(%) | 78.2 | 75.8 | 77.5 | **75.3** |

Table 19: Ablation study of Stylized-ImageNet (SIN), SelfNorm(SN), CrossNorm(CN), and cropping with ResNet50 trained 90 epochs on ImageNet. Compared with SIN, CN holds a better balance between clean and corruption errors. SN also works well in decreasing the corruption error while maintaining low clean error. Combining SN and CN can further lower both errors.

| | Basic | SIN | SN | CN | SNCN | SNCN+Crop |
|---|---|---|---|---|---|---|
| Clean error (%) | 23.9 | 27.2 | 23.7 | 23.4 | **23.3** | **23.3** |
| mCE(%) | 80.6 | 73.3 | 73.4 | 75.3 | **70.4** | 71.1 |

Table 20: Ablation on GTAV-Cityscapes domain generalization. Mean IoU for both within-domain and cross-domain evaluation are reported. All methods use FCN with ResNet50 as backbone network. We use style-only crop for segmentation.

| Methods | FCN Baseline | SN | CN | CN+Crop | SNCN | SNCN+Crop |
|---------|--------------|------|------|---------|------|-----------|
| Source  | 63.7         | 64.6 | 61.8 | 61.2    | **65.0** | 63.5  |
| Target  | 21.4         | 29.9 | 31.5 | 32.0    | 34.1 | **36.5**  |

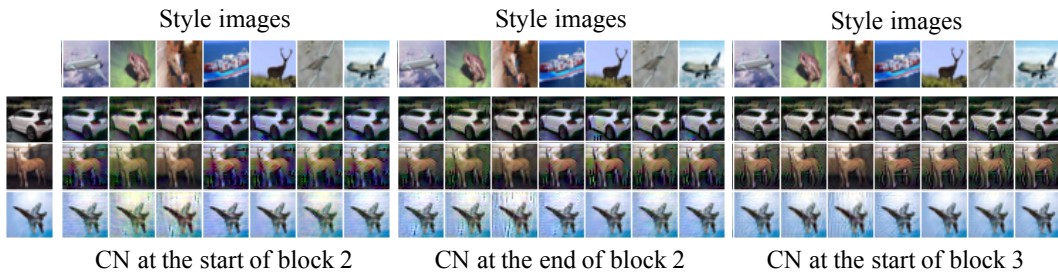

| Style images | Style images | Style images |
|---|---|---|
| CN at the start of block 2 | CN at the end of block 2 | CN at the start of block 3 |

Figure 11: CrossNorm visualization at the head (**Left**), the tail of (**Middle**) block 2 and the start of block 3 (**Right**) in a WideResNet-40-2. Both the content (**Row**) and style (**Column**) images are from CIFAR-10. Compared to CrossNorms in block 1, shown in Figure 5, the CrossNorms in blocks 2 and 3 have weaker style transfer effects. Because the channel-wise means and variances in high-level feature maps may contain less low-level visual information.

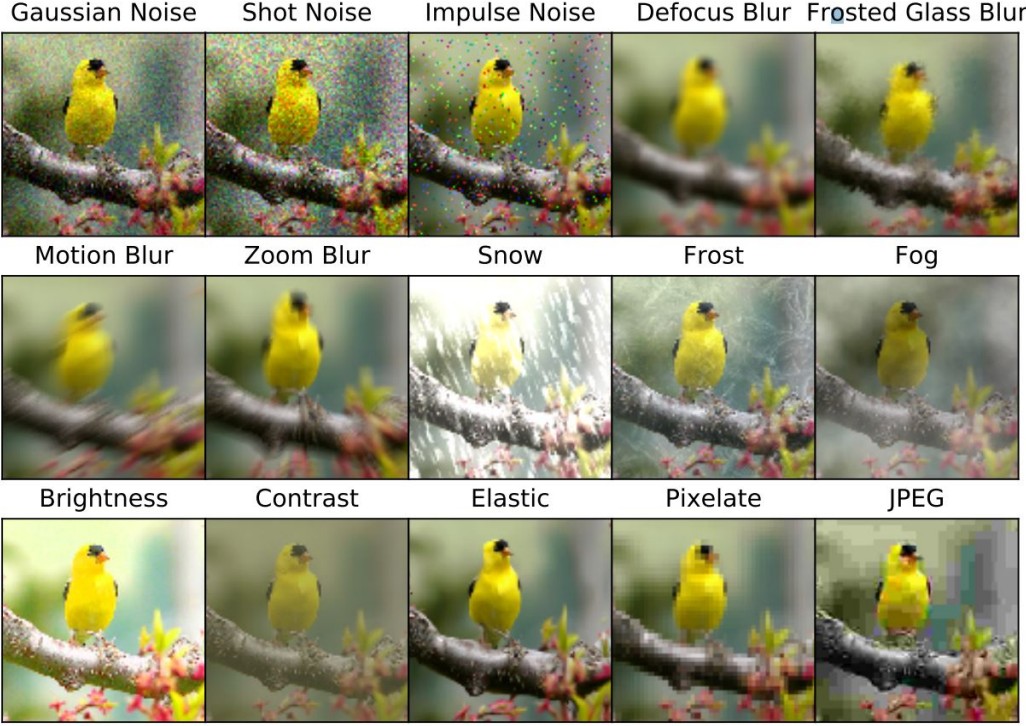

Figure 12: A demostration of corrupted images in ImageNet-C dataset (Hendrycks & Dieterich, 2019). 15 types of algorithmically generated corruptions from noise, blur, weather, and digital categories are applied to images to create corrupted dataset.

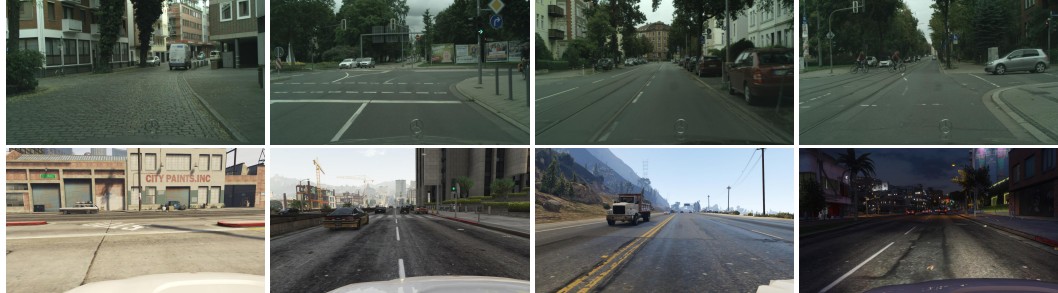

Figure 13: Some examples of segmentation dataset. The first row are images from Cityscapes dataset, while the second row are images from GTA5 dataset.

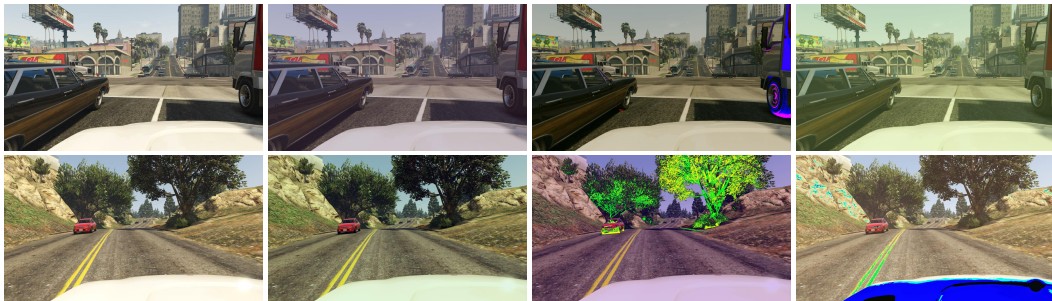

Figure 14: A visualization of CrossNorm with crop style used on image level. The two images on the first column are the original images in the GTA5 dataset and are in the same training batch. We applied CrossNorm to these two images several times and got the following three pairs of images. By calculating style from random crops, CrossNorm can perform a variety of style augmentation.It should be noted that we use CrossNorm in both image and feature level, we only visualize in image level here for simplicity.

