# OpenReview forum: "SelfNorm and CrossNorm for Out-of-Distribution Robustness"
_ICLR.cc/2021/Conference — Reject_

### Official Review · AnonReviewer1 · 2020-10-19
**Not sure yet if the paper can be published.**

**Rating:** 5
**Confidence:** 4

**Review:**

**Summary**
The paper introduces two simple modules, SelfNorm (SN) and CrossNorm (CN), that are highly modular and can theoretically be attached to different parts of the CNNs to control the balance between style and content cues for their recognition. SN is a lightly parametrised module that gives freedom to the CNN to dynamically re-adjust the means and variances of the feature maps in the original instance normalization framework. The authors argue that SN then learns to emphasize _important styles_ and suppresses less important ones, with the underlying assumption that the first and second-order statistics are often sufficient and necessary representations for style. CN is a twin invention that randomly swaps the mean and variance statistics of features of two channels. This leads to diversified virtual styles in the training data, effectively factoring out the model's dependence on style cues for recognition, according to the authors.

**Pros**
It is great news to the field that such simple recipes introduce gains in the performances. From the industrial point of view, the expansion of choices for model design means the likelihood of introducing uniform gains across the board in industrial applications (think about how much $ worth it will be if it cuts the error rate by 0.1 pp across all image classifier applications). From the research point of view, it is exciting to confirm yet again in 2020 that there is much room for improvement in those modular components, after the introduction of SE blocks and the likes.

**Cons**
The above excitement should be checked with a pinch of salt. I find it difficult to follow the rationale behind key assumptions (e.g. that shape = content and texture = style) and the experimental section is, I have to admit, a bit chaotic. And I suspect if the chaos is intended, for downplaying the discouraging side of the results.

1. **Shape=content and texture=style? No.**
First of all, we need a decent, if not mathematical, definition of the four terms above. To me, "content" is the causal cue that constitutes the GT label for the task at hand; I refer to all the rest cues as "style". "Texture" to me is a local pattern that can be captured by sliding windows of, say, size 10x10 pixels and "shape" is any pattern that is more global than texture. Under this terminology, I find it hard to agree that texture is not content. Texture does contribute, as the causal cue, to the final task at hand in many real-world computer vision applications - texture classification, fine-grained cat categorization, medical diagnosis with CT scans, and semantic parsing for detecting snowy and watery road conditions, to name a few.

2. **The oversimplistic view of the world that underlies this paper makes the justifications for the proposed methods all the more fragile.**
It is difficult to agree that SN and CN, which are argued to control style and content for the benefit of the recognition task at hand, are really working as speculated. It is also disputable whether the first and second-order statistics really encode the style and/or texture component. I find it a bit dangerous to let such a convenient and simplifying view of the matters be published and guide researchers in the field to adopt the same kind of viewpoint. In the revision, try to introduce more depths in your arguments and include more empirical analysis to make a point SN and CN work in the promised way.

3. **Introduce order in the experimental reports.**
So I made a table below to summarize the experimental setups in this paper (sorry for the dots everywhere - formatting tricks ;)). I'm having a hard time convincing myself that SN and CN really work empirically, given those highly specific choices of settings per task and analysis. For example, why is ImageNet performance for SN and CN not compared against augmentation baselines considered in the CIFAR experiments in Tab1? What are the individual performances of CN and SN for CIFAR on those 4 architectures in Tab1? What is the effect of location for SN in a CNN (equivalent analysis for CN is presented in Tab4)? There are many, many questions unanswered. I do observe a few improvements introduced by the two modules here and there, but I can't forgo the impression that these are only selected highlights that comply with the authors' arguments. Please introduce the much-needed order in the experimental section, and only then will I be able to assess if the new technology is truly innovative.
| .  Section  .  | .  Data  . | .  Arch  . | .  Evaluation  . | .  Baselines  . | .  Authors' methods  . |
| -- | -- | -- | -- | -- | -- |
| .  Tab1  . | .  CIFAR  . | .  4 archs  . | .  mCE,CleanAcc  . | .  Cutout,Mixup,Cutmix,AA,Advtr,AugMix  . | .  SNCN, SNCN+AugMix |
| .  Fig2  . | .  CIFAR  . | .  28-2WideResNet  . | .  mCE,CleanAcc  . | .  WA,RA  . | .  CN |
| .  Fig4  . | .  CIFAR  . | .  40-2WideResNet  . | .  mCE  . | .  VanillaModel  . | .  SN,CN |
| .  Tab4  . | .  CIFAR  . | .  40-2WideResNet  . | .  mCE  . | .  VanillaModel  . | . CN |
| .  Tab5  . | .  CIFAR  . | .  40-2WideResNet  . | .  mCE  . | .  VanillaModel  . | . SN,CN,SNCN,SNCN+Crop,SNCN+Crop+CR |
| .  Tab2  . | .  ImageNet  . | .  ResNet50 . | .  mCE,CleanAcc  . | .  PU,AA,MaxBlur,SIN,AugMix  . | . CN,SN,SNCN+AugMix |

**Key reasons for the rating**
The simplicity and effectiveness of the technologies argued by the authors are eclipsed by the oversimplifying assumptions and inefficient experimental exposition. The rating reflects this disappointment. Please aim to improve the paper in the rebuttals and paper revisions.

---

> ### Author Response · Authors · 2020-11-25
> **Thank you for your feedback and suggestions.**
>
> We appreciate your feedback and suggestions and we made substaintial modifications in the updated version. As for your concerns:
>
> **Concept**:
>
> 1. *"Shape=content and texture=style? No":*
>
>  We agree that content and style are more general concepts than shape and texture. To avoid confusion, we have substantially rewritten our paper and remove the concepts of shape and texture. We also give a tentative definition of style in the introduction section. Theoretically, content and style can refer to causal and irrelevant and cues that constitute GT labels. In practice, it is not easy to separate them from raw data. Going from hand-crafted features, e.g., HOG, to learned deep features, research is still exploring better feature representations. In light of this, we define the style in practice as a family of weak cues that may connect to interest content. For object recognition, style may refer to appearance-related factors such as color, contrast, and brightness.
> That is to say, some style cues may still be useful sometimes, but the cues as a whole are not as important as other more content-related information such as object shape. Moreover, content and style may have different instantiations from task to task. For example, image texture is content in texture classification, whereas it belongs to style cues in object recognition.
>
> **Assumption:**
>
> 2. *"The oversimplistic view of the world that underlies this paper makes the justifications for the proposed methods all the more fragile.":*
>
>  We agree that using the channel-wise mean and variance to represent style seems to be simplistic. Nevertheless, the simple representation has been widely used in the style transfer field [1, 2]. We have revised our paper to address the assumption concern. Please refer to the last paragraph in the introduction section regarding the assumption. We clarify that the mean and variance are only one way, probably not the best one, to represent style. But they are simple yet effective. In this revision, we also provide more visualizations and empirical analysis of SN and CN in the introduction and experiment section 4.4. We can find that SN can suppress trivial styles, and CN can generate diverse style augmentations.
>
> **Experiments:**
>
> 3. *"Why is ImageNet performance for SN and CN not compared against augmentation baselines considered in the CIFAR experiments in Tab1?":*
>
>  We use the same baselines as those in Tables 1 and 2 of AugMix [3]. We follow AugMix's setting because it sets benchmarks on different architectures and datasets and has public code: https://github.com/google-research/augmix. Please note that the released code trains on ImageNet for 90 epochs, different from 180 epochs in the AugMix paper. For a fair comparison, we train all models 90 epochs on ImageNet in our experiments.
>
>  Also, we split SelfNorm and CrossNorm in the ImageNet comparison (Table 2) because we need to compare CrossNorm to Stylized-ImageNet (SIN). Since our CrossNorm (CN) is closely related to SIN, readers may be interested in their comparison. To make our methods in Table 2 more consistent with Table 1, we have added the SNCN variant in Table 2 and moved the CN v.s. SIN comparison to Table 19 in the appendix.
>
> 4. *" What are the individual performances of CN and SN for CIFAR on those 4 architectures in Tab1?":*
>
>  Please refer to Tables 8-15 in the appendix. The results show that SN and CN can contribute to the robustness separately and jointly on 4 architectures. We provided them as supplementary materials due to the limited space in the main body of the paper.
>
> 5. *"What is the effect of location for SN in a CNN (equivalent analysis for CN is presented in Tab4)":*
>
>  Table 6 in the appendix shows the results of SN and CN on block choices. They both work the best when plugged into all blocks. We did not test different block choices of SelfNorm because we follow the popular SE-Net [4] routine that plugs a SE attention module into each residual module. We did the ablation study for CrossNorm as there is no experience we can draw on from previous research.
>
>
> &nbsp;
> &nbsp;
> &nbsp;
> &nbsp;
>
> [1] Arbitrary style transfer in real-time with adaptive instance normalization.  ICCV 2017.
>
>
> [2] Instance normalization: The missing ingredient for fast stylization. Arxiv 2016.
>
> [3] Augmix: A simple data processing method to improve robustness and uncertainty. ICLR 2020.
>
>
> [4] Squeeze-and-excitation networks. CVPR 2018

---

> > ### Comment · AnonReviewer1 · 2020-11-25
> > **Thank you for the response.**
> >
> > Thank you for the detailed response and paper revision. I have read the other reviewers' comments as well as the revision and the authors' responses.
> >
> > 1. On concept "Shape=content and texture=style? No".
> >
> > The authors have addressed the concern to some degree. They have updated the introduction to include more depth to the previously simplistic argumentation. I like R4's request to include more visualization and analysis in the main paper. The authors have done so and it seems to work. The content and style argument is still weak, but at least corroborated by the experimental analysis.
> >
> > 2. On assumption that first and second-order statistics encode style.
> >
> > As the authors have argued, much literature on stylization adopts this viewpoint. Popularity does not always imply correctness, but I do understand that it is unfair to evaluate their argument with a higher bar than what is widely adopted. Given this, it is commendable that the authors have included a bit of discussion around the validity of the assumption in the intro. I consider this as a good reconciliation.
> >
> > 3. Experiments.
> >
> > Sorry. I really did my best to validate that SN, CN, and SNCN are all useful components, but I'm still not sure. I spent an hour comparing **16 tables** in the original paper and the **20 tables** in the revised paper. The appendix part has become even more chaotic after the revision. At the very least, please write down the performances of SN-only and CN-only in Tables 1 and 2. Initially, they were only included in Table 1 (that's why I complained), but instead of adding the results in Table 2, the authors have *removed* those results from Table 1. In general, I wish the authors be more careful with selecting a different set of methods to compare for different benchmarks. This is often a strong indication that the authors are hiding something. The authors have been unable to clear off this doubt this time. Since I can't spend another hour looking for those results in Appendix (most likely general readers will also have a hard time here), I have to conclude here that this particular concern has not been addressed yet.
> >
> > * "Unity of opposites" - do we need to describe the method this way?
> >
> > Sorry - this is not a point raised by myself in the first round, but still is a point that has confused all the reviewers, as far as I can see. When one says "opposite", it really means the two have opposing characters - like +1 and -1, or like day and night, or like hot and cold. Can you say the same for SN and CN? I don't think so - and this is why all the reviewers got confused. I believe the "unity of opposites" terminology hinders the understanding more than it helps. Maybe consider deleting it?
> >
> > * Conclusion
> >
> > The authors have done great work addressing the concerns. But even with all that, I find it difficult to raise the score without an assurance that the modules (SN, CN, SNCN) are really useful in practice. I did my best to find the supporting evidence in experiments, but I ran out of time. Not sure if there's another chance for the authors to respond, but if they do, please clean up the tables and show the requested results.

---

> > > ### Author Response · Authors · 2021-01-14
> > > **Post-decision responses**
> > >
> > > We are sorry for not being able to respond earlier because the rebuttal deadline has passed. Despite the decision made, we would like to clear two minor concerns.
> > >
> > > 1. Experiments
> > >
> > > Based on the above responses, multiple experiments' concerns have been resolved, and only one seems unclear that SN-only and CN-only do not show up in Tables 1 and 2. As we have pointed out in earlier responses, SN and CN's individual results are given in Tables 14, 15, and 19 of the appendix. We are sorry that too many tables exist in the appendix. We provide all the tables because we want to demonstrate the effectiveness of SN and CN thoroughly with different datasets (CIFAR-10, CIFAR-100, and ImageNet), backbones (AllConvNet, DenseNet, WideResNet, and ResNeXt), cropping options (neither, content, style, and both), and combinations (the consistency regularization and AugMix). This comprehensive exhibition is quite the opposite of hiding something. Therefore, we think this should not be considered as a concern, yet this serves as a major argument leading to the paper decision.
> > >
> > > 2. ""Unity of opposites" - do we need to describe the method this way?"
> > >
> > > Thanks for the discussion. We agree that the way of "+1" and "-1" opposing each other is easier to understand. SN and CN oppose each other in the sense that SN decreases style variance while CN increases style variance. On the other hand, they form a unity because they share the same goal: out-of-distribution robustness. Please note that we have followed R4's advice to remove the term "Unity of opposites" from the title. We keep it in the method section to help readers understand their relationships better as the term can provide some insights.
> > >
> > > In summary, our experiment results are solid and promising, which was initially recognized by the second paragraph in the first-round review comments. We think the post-rebuttal argument is a bit weak as the reason to reject our paper, but thank you very much for your constructive feedback and the efforts in reviewing our paper, we greatly appreciate it.

---

### Official Review · AnonReviewer4 · 2020-10-27

**Rating:** 7
**Confidence:** 4

**Review:**

1. Summary

The paper presents two new methods to improve corruption robustness and domain generalization: SelfNorm, a way to adapt style information during inference, and CrossNorm, a simple data augmentation technique diversifying image style in feature space. Both methods are tested on Cifar10/100-C, ImageNet-C, Semi-Supervised Cifar and GTA V -> Cityscapes.


2. Strengths
+ The approach is straight forward and apparently very effective
+ The evaluation is performed reasonably
+ The method is tested not only on corruptions but also on domain adaptation. This is great as both tasks can be seen as two sides of the same metal and it is great to see more methods testing on both.

3. Weaknesses
- The main problem is the write up. In the introduction "texture sensitivity" is introduced as a concept but never picked up later on. In general the motivation is very high level drawing a lot on the concepts of style, texture and content but the method itself is rather down to earth modifying instance normalization parameters. In my opinion the high level arguments and colorful terms (style, shape sensitivity, unity of opposites etc.) distract rather than add to the story. They may serve as inspirations but without extensive experiments it is hard to related concepts like style to the manipulation of network parameters.
- The ablation study was hard to follow and tbh I think it could have been shortened presenting only the most important results in the paper and the rest in the appendix.
- Conversely I felt the paper would profit from more figures and visualizations (e.g. Figure 7). The method is very simple and I think that is a major strength. It should thus also be possible to present it in a more concise form.


4. Recommendation

As it is right now I think the paper has to be rejected because the write up is just too chaotic and vague. I do however like the method a lot and I would vote for accept if it was rewritten substantially to focus more on an understandable presentation of the method than abstract concepts and colorful terms.


4. Questions/Recommendations
- It is probably beyond the scope of the review period but a demonstration of the technique on some non image data would be amazing making the theoretical argument in Section 4.1 that the method can work on other domains much more powerful.
- It would be nice to include a discussion of "Improving robustness against common corruptions by covariate shift adaptation".

6. Additional feedback
- Figures 1 and 2 are a bit hard to understand.
- IMHO it would be more interesting to include Figure 7 in the paper and move Figure 3 into the appendix.
- The use of IBN should be mentioned in Table 2

---

> ### Author Response · Authors · 2020-11-25
> **Thank you for your feedback and suggestions.**
>
> Thank you for your feedback and suggestions. We have rewritten the paper significantly by considering all the suggestions.
>
> 1. "*I do however like the method a lot and I would vote for accept if it was rewritten substantially to focus more on an understandable presentation of the method than abstract concepts and colorful terms.*"
>
>  We have carefully considered the suggestion and rewritten the paper substantially. Besides, we would like to add some notes on our paper story. The tool (channel-wise statistics) we use is abstract for our high-level goal (out-of-distribution robustness). Our story is mainly to connect them in an easily understandable way.
>
>  In the former version, we describe our story in a top-down way that starts from the intuitive shape and texture concepts, shown to have close relations to robustness [1]. Then we also follow [1] to further link shape and texture with content and style. Finally, the channel-wise mean and variance come out because they have been widely used to represent style in the style transfer field [2,3].
>
>  Following the suggestion, we have revised our story to a bottom-up form. We start with some illustrative examples, shown in Figure 1, which can directly relate the channel-wise statistics to image appearance changes. The model robustness to appearance changes is a fundamental goal in out-of-distribution generalization, as demonstrated in the current popular datasets [4, 5]. We have tried to avoid using the concepts of shape, texture, and content. We retain the style and provide its tentative definition. Style is necessary because it is widely used in style transfer research, and we need a concise and intuitive word to represent the effect of modifying the statistics. Without it, the story will become wordy and hard to understand, hence impaired quality. Another change is that we have removed "unity of opposites" from the title. We still keep the relation analysis inside the paper to make it integrity.
>
> 2. "*It is probably beyond the scope of the review period but a demonstration of the technique on some non-image data would be amazing making the theoretical argument in Section 4.1 that the method can work on other domains much more powerful*"
>
>  Following the suggestion, we have tested SelfNorm and CrossNorm on sentiment analysis, a NLP task. We train CNN classifiers on IMDb [6], which contains full-length lay movie reviews, and evaluate it on another dataset SST-2 [7], which contains pithy expert movie reviews.  Models predict a movie review’s binary sentiment, and we compare their accuracy. Results in Table 4 show that our methods can also effectively improve out-of-distribution generalization in the NLP domain.
>
> 3. "*It would be nice to include a discussion of "Improving robustness against common corruptions by covariate shift adaptation".*"
>
>  Thanks for bringing this paper to our attention. We have added it in the related work section. It mainly identifies a new setting in improving corruption robustness. Technically, it adapts the Batch Normalization statistics using unlabeled corrupted data in testing. It is related to ours in the sense that both manipulate normalization statistics to improve model robustness. Despite that, they are different in several aspects. The most obvious one is that it is an adaptation method, assuming access to the test data, while ours uses a harder generalization setting without any test information. Besides, ours uses the Instance Normalization statistics in contrast to their Batch Normalization ones.
>
>  Besides, it is interesting to compare this paper's story to ours. Its part of connecting robustness to modifying BN statistics is straightforward. First, it makes an intuitive analogy of clean and corrupted data to source and target data in domain adaptation. Adapting Batch Norm (ABN) is already an existing method in domain adaptation, thus applicable there. The logic is simple mainly because it relies on the well-defined domain adaptation concept and existing ABN. Our story, however, does not have such well-defined concepts to link the low-level normalization statistics to the high-level goal. That is one reason why our story narrates itself differently.
>
>
> [1]. ImageNet-trained CNNs are biased towards texture; increasing shape bias improves accuracy and robustness. ICLR 2019.
>
> [2]. Arbitrary style transfer in real-time with adaptive instance normalization. ICCV 2017.
>
> [3]. Instance normalization: The missing ingredient for fast stylization. Arxiv 2016.
>
> [4]. Benchmarking Neural Network Robustness to Common Corruptions and Perturbations. ICLR 2019.
>
> [5]. Playing for data: Ground truth from computer games. ECCV 2016.
>
> [6]. Learning word vectors for sentiment analysis. ACL 2011.
>
> [7]. Recursive deep models for semantic compositionality over a sentiment treebank. EMNLP 2013.

---

> > ### Author Response · Authors · 2020-11-25
> > **Responses continued**
> >
> > 4. "*Figures 1 and 2 are a bit hard to understand.*"
> >
> >   In the revision, we have simplified previous Figures 1 and 2 to make them easier to understand.
> >
> > 5. "*IMHO it would be more interesting to include Figure 7 in the paper and move Figure 3 into the appendix.*"
> >
> >   Thanks for the suggestion. We provide a new section 4.4 to give comprehensive visualizations of SelfNorm and CrossNorm together with their analysis. We put previous Figures 3 and 7 both into the appendix due to the space limit.
> >
> > 6. "*The use of IBN should be mentioned in Table 2.*"
> >
> >  Thanks for pointing it out. We have updated Table 2's caption to reflect it.
> >
> > 7. "*The ablation study was hard to follow and tbh I think it could have been shortened presenting only the most important results in the paper and the rest in the appendix.*"
> >
> >  Thanks for the suggestion. We have moved all ablation studies to the appendix. At the same time, we provide additional experimental results in the NLP domain (Section 4.3) and more visualizations (Section 4.4).

---

### Official Review · AnonReviewer2 · 2020-10-28
**he paper contributes a solution by forming a unity of opposites in using style for model robustness. Both the motivation and intuition are clearly presented. The idea is somewhat novel for me.**

**Rating:** 7
**Confidence:** 4

**Review:**

This paper investigates the CNN model robustness against problems, e.g., texture sensitivity and bias. In particular, this paper proposes to recalibrate style using SelfNorm motivated by the fact that attention help emphasize essential styles and suppress trivial ones and reduce texture bias using CrossNorm by swapping feature maps within one instance. The opposite two processes forms a unity of using style to advance model robustness. The intuition of the proposed method is clearly presented and the paper is well structured. Detailed comments are summarized as follows:

Pros:
	- The paper contributes a solution by forming a unity of opposites in using style for model robustness. Both the motivation and intuition are clearly presented. The idea is somewhat novel for me.
	- The method can be easily implemented and integrated into classical frameworks.
	- Extensive experiments under different settings and tasks show the effectiveness of the proposed method.

Cons:
	- In Sec. 3 Unity of Opposites, the authors explains that SelfNorm works during testing while CrossNorm functions only in training. The statement cannot clearly illustrates the process in training and testing. I suppose both SelfNorm and CrossNorm would be used during training to get a trade-off point and only SelfNorm is used during testing. It is encourage to add an algorithm flowchart to indicate the process.
	- The authors explains SelfNorm recalibrate feature style while  CrossNorm performs style augmentation. It is suggested to visualize the attention/feature maps before and after using the proposed techniques to illustrate their benefits.
The experiment results of SNCN are not significant compared with AugMix though their combination achieves new state-of-the-art.

---

> ### Author Response · Authors · 2020-11-25
> **Thank you for your feedback and suggestions.**
>
> We appreciate your feedback and suggestions and we made some modifications in the updated version. As for your concerns:
>
>
> 1.  *"The statement cannot clearly illustrates the process in training and testing. It is encourage to add an algorithm flowchart to indicate the process.":*
>
>  Thanks for the suggestion. SelfNorm, as a learnable module, needs to learn in training to work in testing. CrossNorm is a data augmentation method, functioning only in training. SelfNorm and CrossNorm target the testing and training stages, respectively. Following the suggestion, we have added a flowchart in the paper as Figure 3 to make it more clear.
>
>
>
> 2.  *"It is suggested to visualize the attention/feature maps before and after using the proposed techniques to illustrate their benefits":*
>
>  Thanks for the suggestion. We have added comprehensive visualization results and analysis in Section 4.4.
>
>
>
> 3. *"The experiment results of SNCN are not significant compared with AugMix though their combination achieves new state-of-the-art":*
>
>  We agree that there is still a performance gap between SNCN and AugMix. However, a critical advantage of SNCN is its domain agnostic property, making it more general and easier to use than AugMix. SNCN uses the channel-wise mean and variance, while AugMix relies on the domain-specific image operations. In the revision, we have added a new experiment in Section 4.3 showing that SNCN can also improve out-of-distribution robustness in the NLP field. AugMix, however, cannot directly apply to NLP tasks.

---

### Official Review · AnonReviewer3 · 2020-10-28
**Interesting norms for model robustness**

**Rating:** 6
**Confidence:** 3

**Review:**

**Summary**

This paper proposes two novel norms: Selfnorm and Crossnorm for model robustness. Selfnorm recalibrates style in features to reduce texture sensitivity. Crossnorm performs style augmentation (swap channel-wise means and variances) to reduce texture bias. The authors also give some discussions on the distinction and connection between SelfNorm and CrossNorm. The proposed norms can be applied to different settings and tasks. Their method shows state-of-the-art robustness performance on both fully and semi-supervised settings, and classification and segmentation tasks.

**Clarity**

- The paper is well organized and easy to read.

**Strengths**

- Contributions clearly stated and validated.
- Comprehensive experiments to show the effectiveness of their method.
- The proposed method is domain agnostic and can be applied to different settings and tasks.

**Weaknesses**

- SelfNorm is a learned normalization, thus I think the statement "they take opposite actions ... at separate stages testing v.s. training ..." (page 1) is confused. Should SelfNorm be used in both testing and training?

- In the experiments, both SelfNorm and CrossNorm units are placed in a ResNet block. What is the order of their placement? It's not clear in the paper.

- It is better to keep the original formula (as shown in Figure 1(a)) in Eq.2 as well so that it is easy for readers to understand.

- The related residual channel attention (CA) [a] should be discussed in this paper. Because the residual channel attention can be also considered as a normalization that recalibrates/rescales features with channel-wise attention. The authors are suggested to give some discussion on the difference between the CA mechanism and proposed Norms. CA is also a unit that is inserted in the ResNet block so that it's interesting to have an experimental comparison as well.

[a] Zhang at al., Image Super-Resolution Using Very Deep Residual Channel Attention Networks, in ECCV'18

---

> ### Author Response · Authors · 2020-11-25
> **Thank you for your feedback and suggestions.**
>
> Thank you for your feedback. We have carefully revised the paper following the suggestions.
>
> 1.  "*Should SelfNorm be used in both testing and training?*"
>
>  Yes. SelfNorm learns in training and functions in testing. Here we mainly emphasize that SelfNorm and CrossNorm target different stages. To make it clear, we have added a dataflow chart for illustration in Figure 3.
>
> 2. "*Order of SelfNorm and CrossNorm*"
>
>  We put CrossNorm ahead of SelfNorm, as shown in the flowchart (Figure 2). Following is the ablation study on their order, which has little performance difference. SN->CN: 46.9, CN->SN: 46.6.
>
> 3. "*Update Equation 2 according to Figure 1(a)*"
>
>  Thanks for the suggestion. We have updated Equation 2 as per the advice.
>
> 4. "*Discuss and compare with Channel Attention (CA)*"
>
>  Thanks for bringing the work to our attention. We have carefully read the paper and its code for channel attention (CA). We find that CA is similar to the squeeze-and-excitation (SE) module, modeling the interdependence between feature channels. In contrast, our SelfNorm deals with each channel independently. Moreover, SelfNorm recalibrates the channel-wise mean and variance while CA/SE adjusts the channel features directly. Considering the similarity between CA and SE, our analysis for SE also applies to CA. Thus, we put CA and SE together in the related work section. One small difference we find between CA and SE is in their implementations. CA turns on bias in the fully connected layer, whereas SE sets the bias to False. Our experiments show similar performance between the two implementations. SelfNorm is more effective than them in improving the corruption robustness (lowering the corruption error). SE: 51.0, CA: 50.8, SN: 47.4.

---

### Author Response · Authors · 2020-11-25
**Rebuttal summary**


Dear reviewers and area chair,

We want to thank you for the constructive feedback. We appreciate the comments that the idea is novel (R2); the method is simple and effective (R4); the contributions are clearly stated and validated (R3); the evaluation is performed reasonably (R4) with comprehensive experiments (R2, R3); and the improvements are exciting (R1).  We are also grateful that reviewers think the paper is well-organized and easy to read (R3), and both the motivation and intuition are clearly presented (R2).

Moreover, we have made substantial revisions considering all the suggestions. We want to point out three main changes:

1. We have substantially rewritten the paper to address the concerns regarding concepts and assumptions (R1, R4).
2. We provide comprehensive visualizations and analysis in Section 4.4 to understand SN and CN better (R2, R4).
3. We add a new NLP experiment in Section 4.3 to demonstrate the domain-agnostic property of SN and CN (R4).

We hope that this has addressed your concerns. Please let us know if you have further questions.

---

### Decision · Program_Chairs · 2021-01-07
**Final Decision**

**Decision:**

Reject

**Comment:**


 This paper proposes two mechanisms, SelfNorm (used during training and inference) leveraging an attention-based recalibration of mean and standard deviation for instance normalization, and CrossNorm which performs cross-channel swapping of mean/stdev. Is is shown that the combination (often combined with AugMix) performs well across several datasets in terms of model robustness. Overall the paper has strength in the fact that the method is interesting, simple to implement, and modular. However, reviewers brought up a number of issues including the overstated motivation/writing, lack of clarity, and most importantly need for clear experimental results (comparing to uniform/standard baselines) and identification of the separate mechanisms. It is especially uncertain why it is necessary that they are used *together* (often with AuxMix as well) to obtain the strong performance. As a result, the score for this paper is borderline, tending towards a weak acceptance.

It is appreciated that the authors provided a lengthy rebuttal, including new results in a different domain (NLP); however, the reviewers agreed that not all of the concerns were addressed.  After a lengthy discussion, all of the reviewers agree that while the method is simple, modular, and effective when combined (hence the positive scores from some reviewers), the authors fail to describe the underlying reason for the method's gains, especially with respect to the individual parts (SelfNorm vs. CrossNorm) and why the results only come when these rather independently derived modules are used together. The exposition of the experimental results, with differing baselines/conditions that make it very hard to understand where the effect is coming from, exacerbates this issue.

As a result of these concerns, I recommend rejection of this paper. However, the method is interesting and results promising, so I hope that the authors can clarify the writing and improve the presentation of the results (specifically separating out the effects of SelfNorm and CrossNorm, as well as analyzing how they interact together to improve results) and submit to a future venue.

---

> ### Author Response · Authors · 2021-01-14
> **Responses to the final decision**
>
> We are sorry to hear the decision, but we appreciate the area chairs' and reviewers' efforts in discussing the paper. Below, we would like to clarify again the arguments that led to the decision.
>
> 1. "overstated motivation/writing, lack of clarity"
>
> The concerns raised by R4 and R1 regarding concept and assumption have been carefully addressed in our paper revision and rebuttal. R4 has increased the rating from 5 to 7, indicating the clearance of concerns. R1 gave detailed responses saying the concerns are largely resolved by the re-written introduction with added discussions, new visualization experimental results, and widely adopted practice. We are surprised that the addressed concerns are still present in the final decision.
>
> 2. "comparing to uniform/standard baselines"
> "The exposition of the experimental results, with differing baselines/conditions that make it very hard to understand where the effect is coming from, exacerbates this issue."
>
> R1 raised this concern in the initial review, and we have pointed out in our rebuttal that we follow previous published work AugMix and use the same baselines in our Tables 1 and 2 as Tables 1 and 2 in AugMix. R1 did not mention this concern in the post-rebuttal responses, hence a solved concern.
>
> 3. "identification of the separate mechanisms"
> "the authors fail to describe the underlying reason for the method's gains, especially with respect to the individual parts (SelfNorm vs. CrossNorm) and why the results only come when these rather independently derived modules are used together."
> "specifically separating out the effects of SelfNorm and CrossNorm"
>
> First, this concern expressed by R1 is regarding the missing separate results of SN and CN in Tables 1 and 2. We have made it clear in the rebuttal (items 3 and 4) that the individual results are put in Tables 14, 15, and 19 in the appendix due to the space limitation in Tables 1 and 2. According to the results, SN and CN can work well separately and jointly. Second, we have provided our explanations for SN and CN's gains in the second paragraph of Section 4.1. The visualization results and analysis in Section 4.4 are also useful in understanding SN and CN's mechanisms, which has been confirmed in R1's response 1. Thus, the argument challenging why SN and CN have to work together is hard to stand.
>
> 4. "It is especially uncertain why it is necessary that they are used together (often with AuxMix as well) to obtain the strong performance."
>
> This is probably a misunderstanding of our work. Our method already outperforms many previous approaches, according to Tables 1 and 2. Combining with previous state-of-the-art Augmix to obtain better results indicates our method is orthogonal to it. This property is an advantage of our method rather than a weakness. In our paper, we have emphasized that our method is domain-agnostic in contrast to AugMix, mainly developed for image classification. Our new experiments in the NLP domain (Section 4.3) support our claim.
>
> 5. "as well as analyzing how they interact together to improve results"
>
> We have analyzed SN and CN's relations and why they together can get better results in paragraph 4 of Section 3.
>
> Overall, we believe our work is qualified for ICLR and is worth being accepted.